# Nano pom-poms prepared exosomes enable highly specific cancer biomarker detection

Nan He[1,2], Sirisha Thippabhotla[3], Cuncong Zhong [3], Zachary Greenberg[4], Liang Xu [5], Ziyan Pessetto[6], Andrew K. Godwin [6,7], Yong Zeng[8] & Mei He [1,4✉]

Extracellular vesicles (EVs), particularly nano-sized small EV exosomes, are emerging biomarker sources. However, due to heterogeneous populations secreted from diverse cell types, mapping exosome multi-omic molecular information specifically to their pathogenesis origin for cancer biomarker identification is still extraordinarily challenging. Herein, we introduced a novel 3D-structured nanographene immunomagnetic particles (NanoPoms) with unique flower pom-poms morphology and photo-click chemistry for specific marker-defined capture and release of intact exosome. This specific exosome isolation approach leads to the expanded identification of targetable cancer biomarkers with enhanced specificity and sensitivity, as demonstrated by multi-omic exosome analysis of bladder cancer patient tissue fluids using the next generation sequencing of somatic DNA mutations, miRNAs, and the global proteome (Data are available via ProteomeXchange with identifier PXD034454). The NanoPoms prepared exosomes also exhibit distinctive in vivo biodistribution patterns, highlighting the highly viable and integral quality. The developed method is simple and straightforward, which is applicable to nearly all types of biological fluids and amenable for enrichment, scale up, and high-throughput exosome isolation.

[1] Department of Chemical and Petroleum Engineering, Bioengineering Program, University of Kansas, Lawrence, KS 66045, USA. [2] Clara Biotech Inc., Lawrence, KS 66047, USA. [3] Department of Electrical Engineering and Computer Science, University of Kansas, Lawrence, KS 66045, USA. [4] Department of Pharmaceutics, College of Pharmacy, University of Florida, Gainesville, FL 32610, USA. [5] Department of Molecular Biosciences, University of Kansas, Lawrence, KS 66045, USA. [6] Department of Pathology and Laboratory Medicine, University of Kansas Medical Center, Kansas City, KS 66160, USA. [7] University of Kansas Cancer Center, Kansas City, KS 66160, USA. [8] Department of Chemistry, University of Florida, Gainesville, FL 32603, USA. ✉email: mhe@cop.ufl.edu

Despite the tremendous efforts made in developing cancer biomarkers and liquid biopsy for past decades, only a few (less than 25) cancer biomarkers have been approved by FDA for clinical practice[1,2], such as Estrogen receptor (*ER*), Progesterone receptor (*PR*), *HER-2/neu*, *CA-125*, and *PSA*[3]. Extracellular vesicles (EVs) have been emerging biomarker sources for expanding the landscape of cancer biomarker discovery in promoting cancer diagnosis[4–6], immunotherapy[7,8], drug target and delivery[9]. Growing attention has been focused on the exosome type small EVs (sEVs) and their molecular components (e.g., proteins, DNAs, mRNA and miRNA), which has been found in association with a variety of physiological functions and pathological disease states[10]. Exosome secretion is exacerbated from tumor cells and enriched with a group of tumor markers, as evidenced by increased presence in plasma and ascites from patients in variable cancers[11]. However, currently there is no standardized purification method for processing body fluids which often contain diverse EV types, and obtaining homogeneous exosome populations that are specific to their cellular origin and molecular components is not attainable[12,13]. As evidenced, EVs are living cell-secreted membrane vesicles in multiple subpopulations, including membrane shedding microvesicles (100 nm–1000 nm), endosomal multivesicular body released exosomes (30 nm–150 nm), and apoptotic cellular fragment vesicles (≥1000 nm)[14–17]. Due to such large heterogeneity and majority size overlap between vesicle populations, the consensus has not yet emerged on precisely defining EV subtypes, such as endosome derived exosomes[18] which is highly relevant to the disease pathogenesis. The generic term of EVs is recommended by complying with 2018 guidelines from the International Society for Extracellular Vesicles (ISEV) proposed Minimal Information for Studies of Extracellular Vesicles ("MISEV")[18]. Current purification methods that recover the highest amount of extracellular materials, no matter with the vesicle or non-vesicular molecules, are mainly the precipitation polymer kits and lengthy ultracentrifugation-based (UC) approach[19,20]. Such isolation approach is not scalable and unable to differentiate the exosome populations from different cellular origin or other EV subtypes (e.g., microvesicles and apoptotic bodies), neither free proteins[21] or viruses, in turn, posing a huge concern for studying cancer biomarkers from tumor cells derived exosomes. The bulk measurement of a mixture of vesicle populations could potentially mask the essential biosignatures, which severely impairs the investigations of associated pathological mechanism[22–25]. As the perfectly enriched biomarker sources, using exosomes for mapping multi-omic molecular information specifically to their pathogenesis in cancer biomarker identification is still extraordinarily challenging.

Herein, we introduced an approach using 3D-structured nanographene immunomagnetic particles (NanoPoms), which possesses unique flower pom-poms morphology and photo-click chemistry for specific marker-defined capture and release of intact exosomes from nearly all types of biological fluids, including human blood, urine, cow's milk, and cell culture medium, etc (Supplementary Figs. s1 and s2). Compared to current existing immunomagnetic beads based EV isolation either in small quality or bound to solid surface/particles[26–28], Nano-Poms enable on demand capture and release of intact exosomes, which leads to the expanded identification of targetable cancer biomarkers with enhanced specificity and sensitivity. Zhu etc. also reported the covalent chemistry mediated EV capture/release using click chip for EV purification and diagnosis of hepatocellular carcinoma (*HCC*)[29] and Ewing sarcoma[30]. The trans-cyclooctene (*TCO*) grafted antibody bound EVs with click chemistry motifs are presented after EV release which may introduce alteration of EV surface properties for in vivo applications. We tested in vivo biodistribution of our NanoPoms prepared exosomes after release and results support the retaining

of biological surface properties with distinctive biodistribution patterns specific to exosome cellular origin, which implies the great potential for therapeutic development.

The group of enriched biomarkers carried by exosomes, including DNAs, RNAs and proteins, could offer the unmatched possibility to integrate multi-omic data analysis for expending the landscape of cancer biomarker discovery and precisely defining the onset and progression of cancer diseases[31]. In this paper, we demonstrated such capability for analyzing exosomes derived from bladder cancer patient tissue fluids including urine and plasma, and compared with tumor tissues by the next generation sequencing (NGS) of somatic DNA mutations, miRNAs, and the global proteome, for achieving non-invasive, ultra-sensitive diagnosis of bladder cancer. The results showed improved specificity and sensitivity for detecting urological tumor biomarkers from NanoPoms isolated exosomes compared to other ultracentrifugation or bead isolation approaches. We also identified a few miRNAs and proteome cancer biomarkers highly enriched in urinary exosomes but was not reported by others yet, which could expand the landscape for discovering novel EV cancer biomarkers for improving bladder cancer diagnosis.

## Results

**NanoPoms enable specific capture and on-demand release of intact exosomes.** In this work, we introduce a 3D-structured nanographene immunomagnetic particles (NanoPoms) with unique flower pom-poms morphology and photo-click surface chemistry for specific marker-defined capture and release of intact exosomes (Fig. 1). Conventionally, the non-covalently assembled nano-graphene suffers from the instability in buffer solutions over time[32]. Our method interfaces $Fe_3O_4/SiO_2$ core-shell particles (~800 nm) with graphene nanosheets via carboxamide covalent bonds, which leads to substantially improved stability in the aqueous samples. The flower pom-poms morphology produces the unique 3D nano-scale cavities in between for affinity capture of only nano-sized vesicles such as exosomes (Fig. 1 and Supplementary Fig. s1). The dense nano-graphene polydopamine sheet layers provide much larger surface area[33–36] for immobilization of affinity capture entities (e.g., antibodies, aptamers, and affinity peptides) as shown in Fig. 1b right panel in contrast to conventional beads. Transmission electron microscope (TEM) cross sectional view illustrates the plumose surface morphology, which is further confirmed by the scanning electron microscope (SEM) top view showing the greatly enhanced layers of surface area. The immune gold nanoparticle staining TEM imaging utilized anti*CD63* gold nanoparticles (~10 nm) to double confirm the surface captured exosomes. The insert in Fig. 1c shows the captured single exosomes in the size range of ~100 nm with three gold nanoparticles bound (~10 nm). The exhibited size distribution range is much narrower than ultracentrifugation (UC) prepared EVs (Fig. 1d). The isolated exosomes are uniform and reproducible in particle size around 100-150 nm, which is smaller than ExoEasy prepared EVs (Fig. 1e). Most importantly, the conjugated photo-click chemistry on bead surface allows the release of intact, captured exosomes on demand, which further ensures the specificity for harvesting marker-defined exosome subpopulations. We observed the dense and round-shaped exosomes (~100 nm) completely covering the surface of NanoPoms and subsequently restoring the pom-poms surface morphology after light release (Fig. 1f). The X-ray photoelectron spectroscopy (XPS) analysis, TEM, fluorescence binding analysis and BCA Protein Assay were also performed to evaluate exosome capture performance and capacity in Supplementary Figs. s1, s2 and s3.

**NGS analysis of somatic DNA mutations carried by urinary exosomes.** Detecting DNA mutations carried by urinary tumor

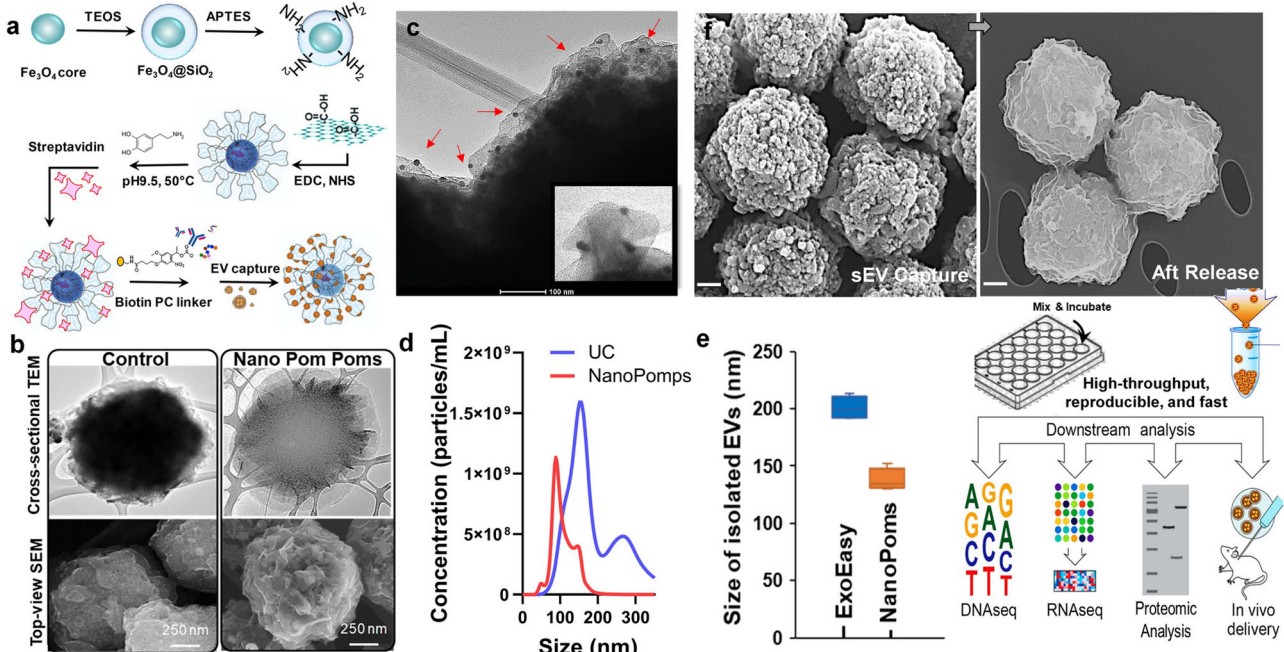

**Fig. 1 Nano pom-poms fabrication for highly specific exosome isolation and multi-omic biomarker analysis. a** Schematic illustration of the fabrication of Nano pom poms. **b** TEM and SEM images showing the unique 3D nano-scale flower pom-poms morphology compared to commercial immunomagnetic beads. **c** The immune gold nanoparticle staining TEM imaging of captured exosomes fully covering Nano pom-poms surface. Captured EVs are confirmed by antiCD63 gold nanoparticles. The insert shows the captured single exosomes in the size range of ~100 nm with three gold nanoparticles bound (~10 nm). **d** Nanoparticle tracking analysis of NanoPoms isolated exosomes with much narrower size distribution in comparison with UC isolated EVs. **e** Nanoparticle tracking analysis of the size of NanoPoms isolated exosomes ($n = 4$ independent experiments, mean ± SD), compared with ExoEasy isolation ($n = 4$ independent experiments, mean ± SD), which showed reproducible and smaller size of exosomes from NanoPoms preparation. **f** SEM images showing the dense exosomes are captured covering the surface of Nano pom-poms, and can be completely released via on-demand photo-cleavage. After release, intact exosomes can be harvested for downstream multi-omic analysis including next generation sequencing of DNAs, RNAs, western blotting and proteomic analysis, as well as in vivo study.

exosomes is emerging, yet challenging, due to the needs of highly pure sample preparation for detection. We analyzed the bladder cancer (BC) patient urine samples prepared by both NanoPoms, UC, and commercial bead approaches for isolating urinary exosomes, with control group from healthy individuals. The NGS GeneRead AIT panel was used to identify the most cancer relevant 1411 variants. UC preparation was found insensitive to cancer relevant variant detection, as it requires much larger urine sample input (4 mL) with more than 100 ng exosome DNAs to give detectable variant signals (Fig. 2a). We suspect that UC isolated exosome DNAs contain more genes which are not specific to cancers. The *PDGFRA* gene (variant c.1432 T > C, *p.Ser478Pro*) with 56.8% detection frequency was observed from a healthy individual in the control group using UC preparation, but not from NanoPoms preparation. NanoPoms prepared healthy control samples did not show any pathological variants. In the BC disease group, NanoPoms prepared exosomes enabled much enhanced detection sensitivity and specificity to BC relevant mutations including *KRAS*, *PIK3CA*, and *ERBB2*, which only consumed 1 mL urine sample with about 10-50 ng exosome DNAs. However, commercial bead isolated exosomes using the same input of urine samples did not yield sufficient DNAs for sensitive detection of cancer relevant variants (Fig. 2a). In order to validate whether the gene mutations found in urinary exosomes are from the urological tumor, we evaluated the matched patient tumor tissue. The NGS GeneRead analysis of tumor tissue cells showed the consistent mutations of *KRAS* and *ERBB2* as carried by urinary exosomes from the same BC patient. Although as one might expect, more mutations were detected in the tumor tissue, including *MTOR* and *BRCA1*; however, the pathogenic

*PDGFRA* variant (c.1939A > G, *p.Ile647Val*) was found in the urinary exosomes from both UC and NanoPoms preparations, but not in the tumor tissue cells (Fig. 2b). It is worth mentioning that the *PDGFRA* variant (c.1939A > G, *p.Ile647Val*) has been reported as the tumor marker from the bladder urothelial carcinoma and the gastrointestinal stromal tumor[37,38].

We also analyzed urinary exosome-derived DNA mutations using droplet digital PCR (ddPCR) from both UC and NanoPoms preparations. While the NGS method provides a broad coverage of interested gene mutation panels in one run, the ddPCR analysis offers much higher detection specificity and sensitivity, and its rapid and low-cost performance is highly attractive for developing clinical cancer diagnostic assays. Thus, we selected *EGFR* (Thr790Met) and *TERT* (C228T and C250T) for ddPCR analysis which are well reported as the prognostic and diagnostic markers for bladder cancer[39–41], although not sensitively picked up by NGS analysis in Fig. 2a. A total of 30 bladder cancer patient urines were analyzed with 10 healthy individuals as the control group. With the same exosome DNA input (10 μg), *EGFR* (Thr790Met) and *TERT* (C228T and C250T) were both detected in Supplementary figure s3. We observed much higher signal amplitudes from NanoPoms prepared exosome DNAs than that from UC approach (Fig. 2c and Supplementary Fig. s4). The average patients' *EGFR* Wt copy number is 3185.4 (±468.3) from NanoPoms approach, which is 12.8-fold higher than that from UC approach (248.9 (±46.4)) with 3-fold higher mutation detection efficiency (Supplementary Fig. s3). The overall detection signal to base ratio from patient group is statistically higher than that from control group (Fig. 2c), indicating the significant diagnostic value (Fig. 2d) for developing liquid biopsy and non-invasive diagnosis of BC using urinary exosomes from

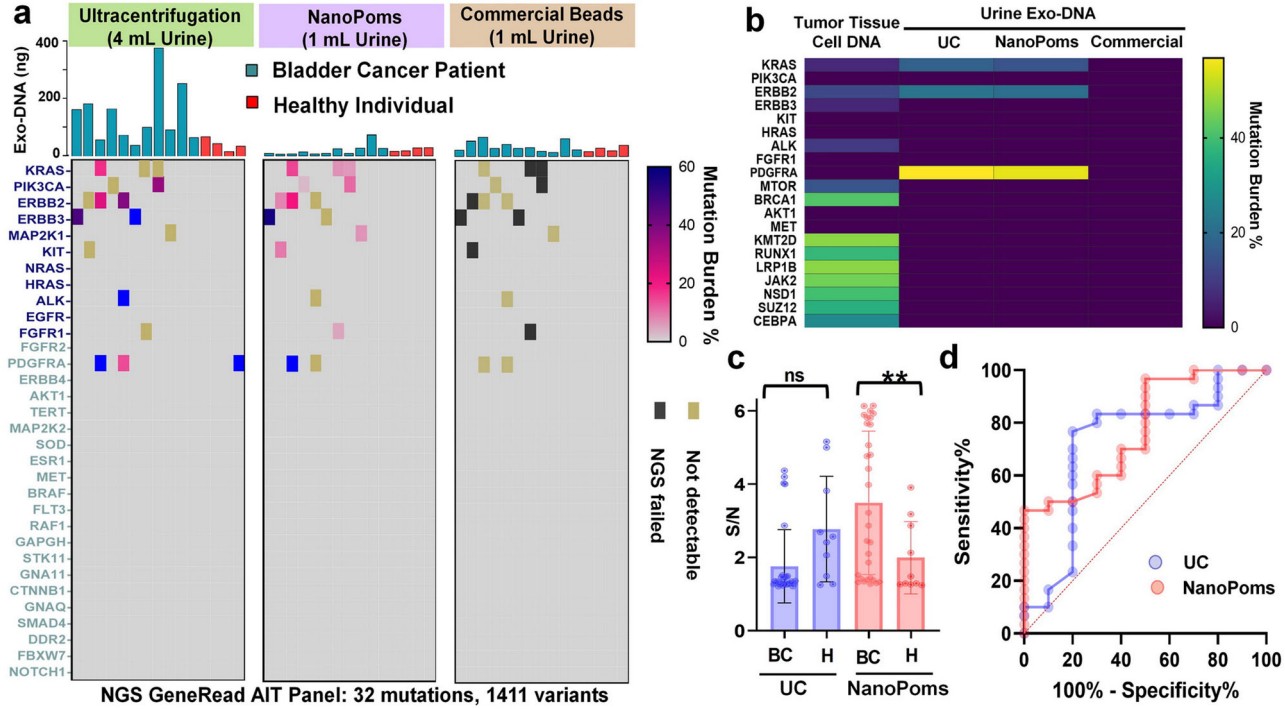

**Fig. 2 The NGS analysis of somatic DNA mutations from bladder cancer patient urinary exosomes. a** The DNA NGS analysis of 11 BC patient urine exosome samples with 4 healthy individuals as the control group using GeneRead AIT panel. Exosomes were prepared in parallel by UC, NanoPoms, and commercial bead approaches to extract total DNAs shown in the bar graphs. The most frequent 1411 cancer relevant variants were sequenced. **b** The NGS GeneRead analysis of tumor cell DNAs from the matched BC patient tumor tissue, compared with urinary exosome DNAs prepared by UC, NanoPoms, and commercial beads. **c** The droplet digital PCR analysis of EGFR (Thr790Met) extracted from purified exosomes using both NanoPoms (pink dots) and UC (blue dots) approaches from bladder cancer patient urine samples ($n = 30$ biologically independent patient samples, mean ± SD) with healthy individuals as the control group ($n = 10$ biologically independent patient samples, mean ± SD). **d** Receiver operating characteristic (ROC) analysis of ddPCR detection of EGFR showing significant diagnostic performance from NanoPoms approach compared with UC preparation. The AUC (Area Under the Curve) for NanoPoms preparation is 0. 78 with $p < 0.01$. The AUC for UC preparation is 0.71 with $p > 0.04$.

NanoPoms preparation. In contrast, UC-based e preparation is unable to differentiate patient group from the healthy control group ($p > 0.05$, Fig. 2c).

Interestingly, we also observed EGFR heterozygous mutation in three BC patients while conducting ddPCR analysis of NanoPoms prepared urinary exosome DNAs (Supplementary Fig. s4). In contrast, UC isolates from the same patients 2 and 3 did not show such heterozygous mutation (Fig. 3a and Supplementary Fig. s4). In order to further validate this observation, we obtained the matched patient plasma and buffy coat with white blood cells (WBC) as the control. NanoPoms preparation allows to pull out marker specific exosome populations based on the exosomal surface markers (*CD9*, *CD63*, and *CD81*) to match urinary exosome populations, which avoids the interferences from other microvesicles or non-disease associated vesicles. Afterwards we used Sanger sequence to confirm the presence of the EGFR heterozygosity for three patients. Results were consistent with ddPCR analysis from NanoPoms preparation (Fig. 3b). As expected, the EGFR heterozygosity was not detected from control WBCs from matched patients. These results clearly support that marker specific capture and release enabled by NanoPoms method can greatly enrich tumor-associated exosomes for sensitive mutation detection. Although the UC preparation yields larger numbers of vesicle particles, their specificity and purity to tumor-associated exosomes are much less than NanoPoms preparation.

**NGS analysis of urinary exosome RNAs**. Analyzing RNAs within urinary exosomes has been emerging with needs for non-

invasive, early detection, and timely medical checkup of BC[42,43]. Exosome long non-coding RNAs (lncRNAs) *PVT-1*, *ANRIL* and *PCAT-1* have been reported as the novel biomarker in BC diagnosis[44–47]. However, NGS profiling of microRNA from tumor derived urinary exosomes from BC patients has not been exploited. In this study, we analyzed urinary exosome microRNA NGS profiles from both BC and healthy individuals.

The distribution of exosome small RNA categories from NanoPoms preparation showed more lncRNAs in both the BC group and healthy control group (42% from NanoPoms vs. 18.9% from UC) (Fig. 4a, Supplementary Table s1). In contrast, UC preparation leads to the higher percentage of tRNA. Although the exact role of exosome lncRNAs is not well understood yet, several studies have showed exosomal lncRNAs are novel biomarkers in cancer diagnosis and are highly associated with cancer progression and cellular functions[48–50]. Currently, only a small number of lncRNAs has been investigated which partially due to the inconsistency imposed by exosome preparation methods[51]. We further look into the top 100 miRNAs expression profiles as shown in Fig. 4b. The heatmap clustering analysis indicates the clear differentiation between BC group and healthy control from NanoPoms exosome preparation, in contrast to UC preparation. We also investigated the influence of photo cleavage process during the exosome harvesting on the integrity of overall exosome miRNAs Supplementary Fig. s5 and did not observe any significant differences or impair on miRNA profiles.

In order to further interpret urinary exosome miRNA profiles and characterize the influences imposed by sample preparation steps, we used the volcano plot to analyze the statistical

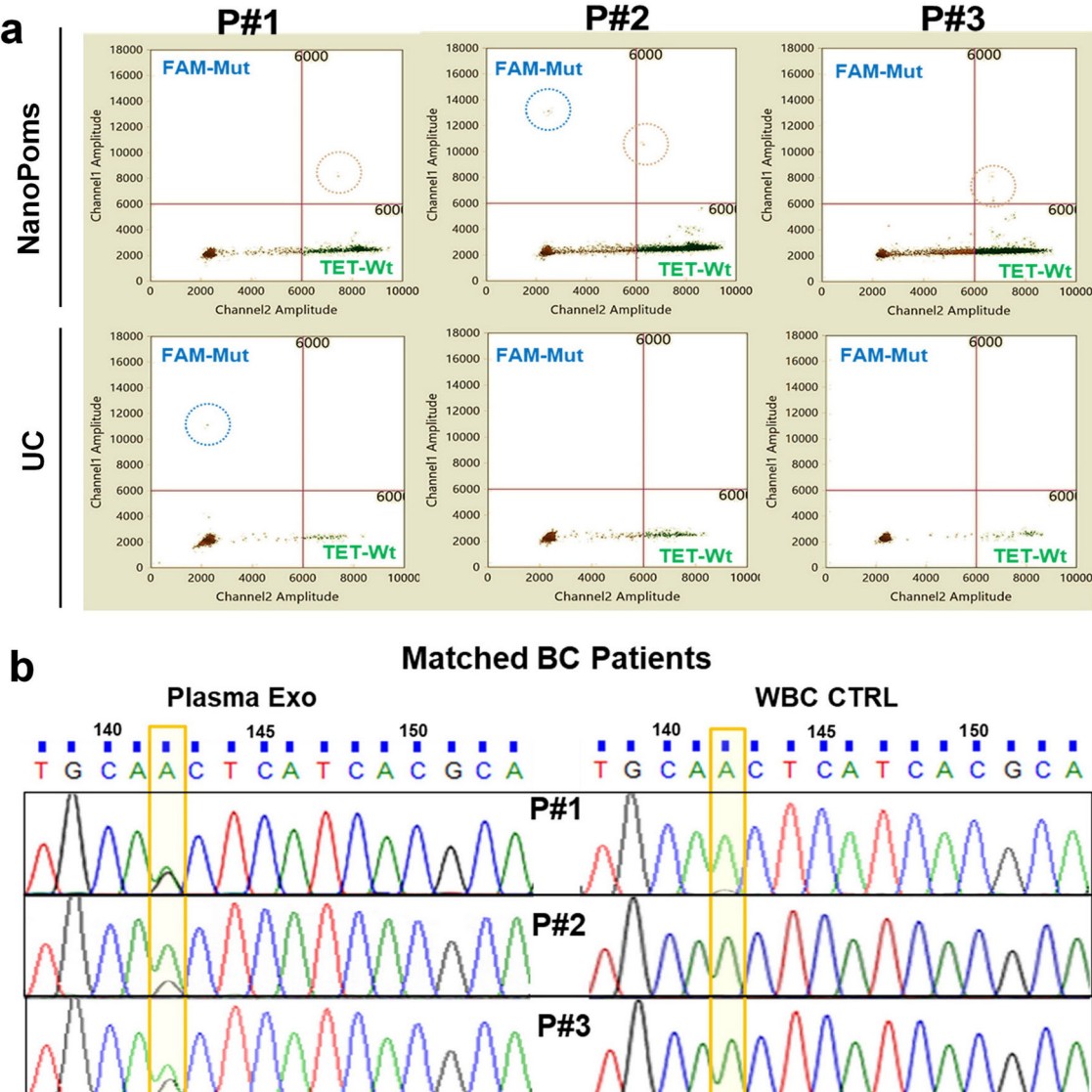

**Fig. 3 NanoPoms prepared exosomes enable highly sensitive detection of heterozygosity using ddPCR. a** The ddPCR detection of EGFR (Thr790Met) heterozygosity from NanoPoms prepared urinary exosomes in three BC patients, compared with UC preparation. **b** Sanger sequence validation of EGFR heterozygosity from NanoPoms prepared plasma exosomes from matched patients in Fig. 3a. Genes from the corresponding patients' white blood cells (WBC) are the wild-type control.

significance (*P* value) versus fold-gene expression changes from both UC and NanoPoms preparations. It is interesting to note that top 10 miRNAs were highly enriched from the NanoPoms preparation, including *hsa-miR-3168*, *hsa-miR-92b-5p*, *hsa-miR-891a-5p*, *hsa-miR-934*, and *hsa-miR-6785-5p* (Fig. 4c and Supplementary Table s2). We searched the reported miRNA functions and found those miRNAs were reported as the cancer relevant markers specifically sorted into exosomes (Supplementary Table s2). For instance, *hsa-miR-3168* has been reported to be enriched in exosomes via a *KRAS*-dependent sorting mechanism in colorectal cancer cell lines[52] and is known as the melanoma mature miRNA[53]. The *miR-92b-5p* has been found to play a critical role in promoting *EMT* in bladder cancer migration[54]. The *hsa-miR-934* is an essential exosomal oncogene for promoting cancer metastasis[55]. NanoPoms exosome preparation offered much higher molecular relevance for identifying tumor associated biomarkers, which is crucial for exploring more specific targetable cancer biomarkers.

**Proteomic analysis of urinary exosome proteins**. The urinary protein biomarkers could enable highly significant clinical values for the cystoscopic evaluations in BC diagnosis. *EDIL-3* (Epidermal growth factor (EGF)-like repeat and discoidin I-like domain-containing protein 3) and mucin 4 (*MUC 4*) both have been reported in exosomes purified from BC patient urines[56,57]. We selected generic exosome markers *CD9*, *CD63*, and *TSG101*, as well as the *EDIL-3* and *MUC4* for Western blotting analysis of urinary exosome proteins prepared by UC and NanoPoms methods, with the human bladder carcinoma cell line *HTB9* as the control.

As shown in Fig. 5a, the Western blot bands from *CD9* across all samples exhibited the consistent intensity, indicating the consistent loading amount of employed protein samples, although the *β-actin* was unable to express from some of our samples as the control. The generic exosomes markers *CD9*, *CD63*, and *TSG101* were consistently expressed in patient urinary exosomes, *HTB9* cells and their exosomes, which indicates

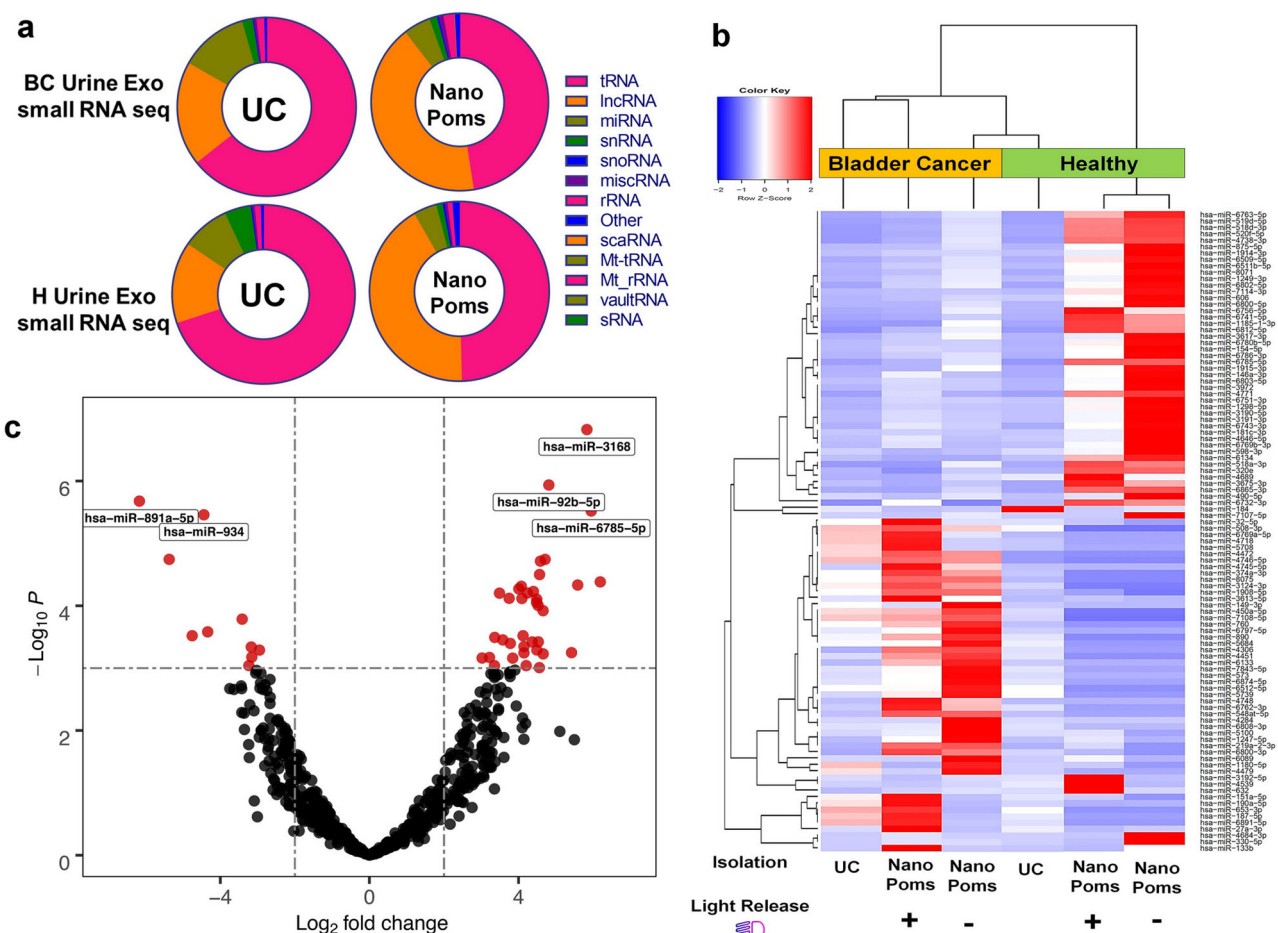

**Fig. 4 The NGS analysis of small RNAs from bladder cancer patient urinary exosomes. a** The distribution of small RNA categories from both NanoPoms and UC prepared urinary exosomes in BC patients and healthy individuals. **b** Heatmap with dendrogram clustering analysis depicts the top 100 highly expressed miRNAs from urinary exosomes isolated from both BC patient and healthy individual using UC, NanoPoms, and NanoPoms without light release process. Red color indicates a higher expression z-score. Hierarchical clustering was performed, using the Spearman correlation method. NanoPoms isolation approach with or without light release processes have been clustered together due to higher similarities in their transcript expressions. **c** Volcano plot analysis depicts the most biologically significant urinary exosome miRNAs with large fold changes identified by using NanoPoms preparation compared to UC preparation. Top 5 highly significant miRNAs are labeled in plot, which are from NanoPoms preparation.

consistent isolation of exosomes. The expression level of *EDIL-3* is much higher in BC patients than healthy individuals, but not in the tumor cell line or their exosomes from conditioned media. *MUC4* protein marker was only observed in the human urinary exosomes and *HTB9* exosomes, but not in *HTB9* cells. This observation supports the previous report that *EDIL-3* and *MUC 4* are highly promising biomarkers in developing urinary exosome-based BC diagnosis and prognosis tests[56]. The proteomic profiling of bladder cancer patient and healthy control urinary exosomes (four biological replications for each group) from NanoPoms preparation was shown in Fig. 5b, and identified proteins were compared with the ExoCarta Exosome Protein Database and the Urinary Exosome Protein Database. Several proteins associated with exosome biosynthesis were observed, such as proteins *PIGQ* and *PAPD7* involved in Golgi apparatus, the cytosol protein *S100-A7* and *A9* found within the exosome lumen which is engaged with natural membrane budding process during multivesicular body formation. We also observed a diverse group of cytosolic enzymes (glyceraldehyde-3-phosphate dehydrogenase) and cytoskeletal constituents (actin, Beta-actin-like protein 2 ACTBL2, and myosin-9). Although the majority of proteins are shared identifications between BC patient and healthy control groups (~82%), as well as the databases we used,

interestingly, we found 11 proteins which are uniquely identified only from BC patient using NanoPoms preparation (Supplementary Table s3). Those proteins have previously been reported to be associated with bladder cancer metastases, including *IRAK4*[58], *KRT23*[59], and *RALGAPA2*[60] (full list in Supplementary Table s3). Also 4 proteins were found uniquely in the healthy group using NanoPoms preparation, but not in cancer patients. From the Human Protein Atlas database (https://www.proteinatlas.org/), those proteins are intracellular and associated with vesicles, Golgi apparatus, and secreted pathway. The identifications are broadly consistent with that expected for exosomes and compatible with other researchers' investigations[61]. Approximately 20% of proteins do not overlapped between the BC patient and the healthy control, which further support the utility of NanoPoms prepared exosomes for diagnosis of BC.

Identified proteins were classified by encoding genes which indicate the majority are located within extracellular exosomes, cytosol, cytoplasm, and the membrane (Fig. 5c). The biological processes associated proteome revealed large associations with the regulation of immune activation, nucleosome assembly, cell organization and biogenesis. The protein binding molecular function from this proteome is dominant. Results exhibit good specificity to exosomal proteome, indicating NanoPoms

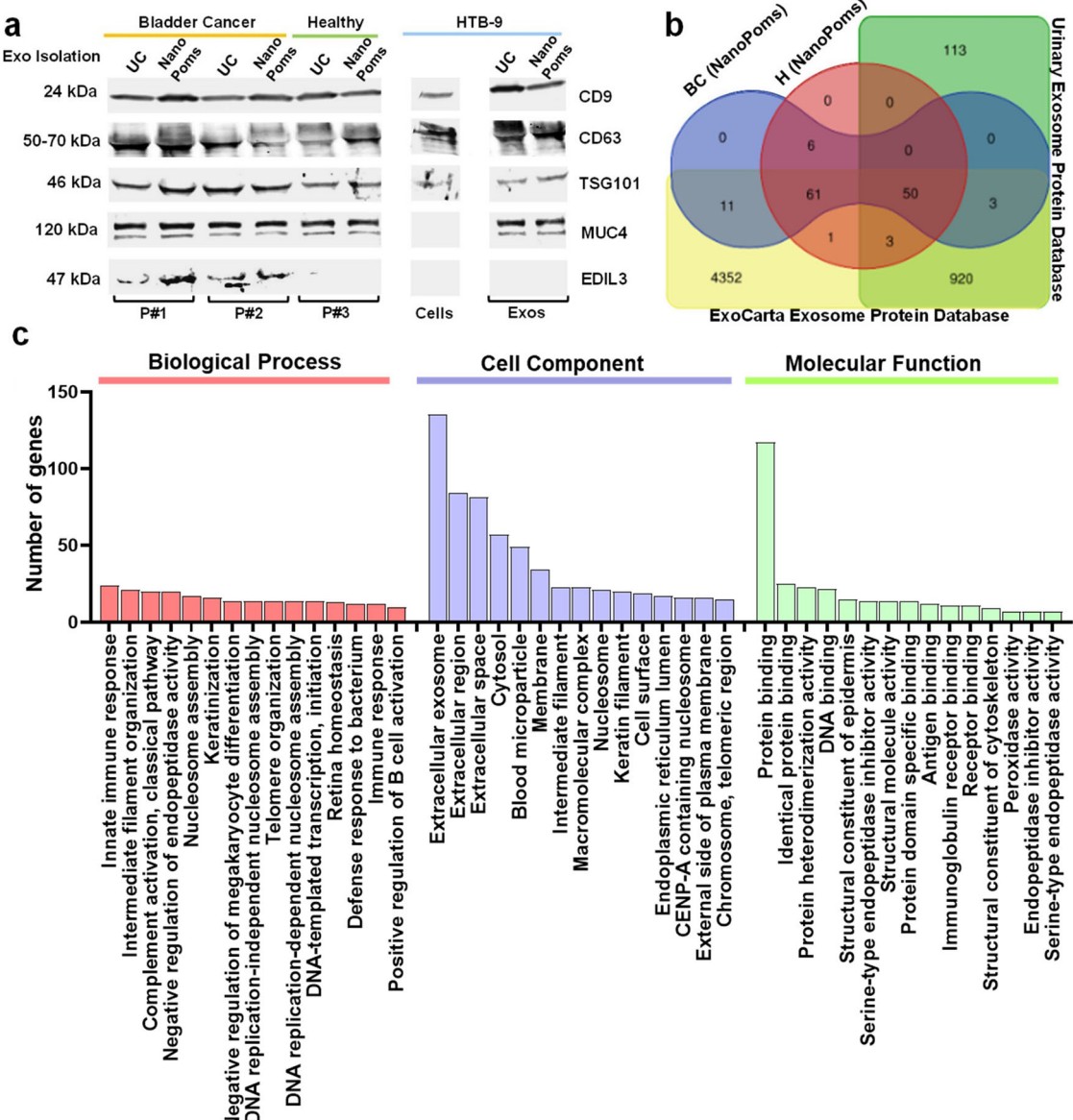

**Fig. 5 The proteomic analysis of bladder cancer patient urinary exosomes. a** Western blotting analysis of urinary exosome proteins prepared by both NanoPoms and UC approaches. Two BC patients and one healthy individual urine samples were used with HTB9 cells and their exosomes from conditioned media as the control. Protein loading amount is applied consistently between samples (~5 μg). **b** Venn diagram illustrates the relationship of proteomes from BC and healthy urinary exosomes prepared by the NanoPoms approach (four biological replications for each group), with references from ExoCarta Exosome Protein Database and the Urinary Exosome Protein Database. **c** Gene Ontology enrichment analysis of differently expressed proteins from NanoPoms prepared BC urinary exosomes. Most abundant items are listed in biological process, cell component and molecular function, respectively.

preparation could provide a pure and high-quality exosome, which could facilitate the important research area in EV proteomics and multi-omics.

**In vivo biodistribution study of NanoPoms prepared exosomes.** The NanoPoms preparation of exosomes via marker specific capture and release is able to collect intact, pure, and homogenous exosome subtypes. Due to the on-demand, light-triggered release process, the molecular engineering, such as the surface modification, drug loading, or dye labeling, can be implemented to immunomagnetically captured exosomes before washing and releasing. This protocol avoids the redundant post purification of small molecules from isolated exosomes, which is often challenging and causes contaminations. For instance, the remaining free dye during in vivo tracking of exosomes could

cause false signals with longer distribution half time, unspecific staining, or tissue accumulation[62]. In this study, we prepared exosomes from bladder tumor *HTB9* cells and non-malignant *HEK* cells with DiR labeling for intravenous tail injection into *BALB/cJ* mice. The buffer solution from beads washing step (without exosomes) was used as the negative control. From these representative images in 24, 48, and 72 hr time intervals post injection (Fig. 6a), organs were harvested and imaged ex vivo in the time intervals of 48 and 72 hr to minimize signal interference (Fig. 6b and c). To rule out of the signal originating from the blood in the organs or from the free dye, we normalized exosome tracking signal with the negative control signal to affirm the in vivo tacking.

In fact, the negative control images did not show much detectable signals indicating no remaining free dye background signal during

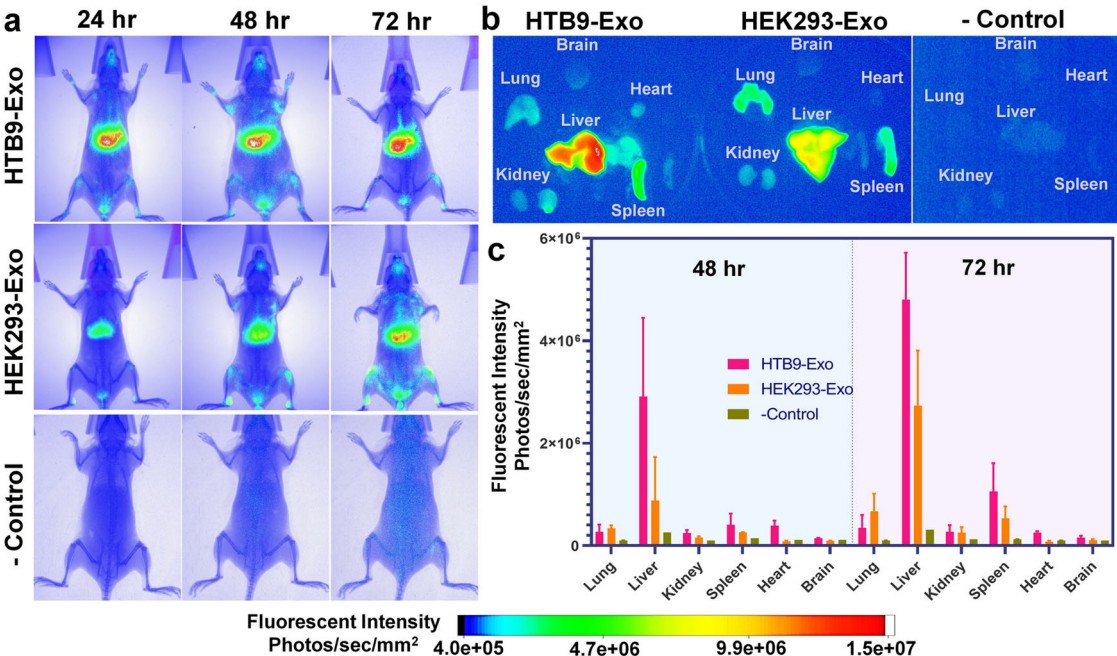

**Fig. 6 In vivo biodistribution analysis of NanoPoms prepared exosomes. a** Representative IVIS images at 24 h, 48 h, and 72 h post-injection of live mice. The HTB9 tumor cell derived exosomes and non-malignant HEK cell derived exosomes with DiR labeling ($2.0 \times 10^9$ particles/ml) were prepared by NanoPoms approach for intravenous tail injection into BALB/cJ mice. The buffer solution without exosomes was used as the negative control. **b** Representative IVIS images of harvested organs (lung, liver, kidney, spleen, heart, and brain) at 48 h and 72 h post injection from mice. **c** The fluorescence signals normalized with negative control from IVIS images in each organ harvested at 48 h and 72 h post injection ($n = 2$ biologically independent animals, mean ± SD).

in vivo tracking of exosomes. By further observing the harvested organs, *HTB9*-derived exosomes exhibit different biodistribution profile in lung, liver, kidney, spleen, heart, and brain, as compared to exosomes isolated from the non-malignant *HEK293* cells. Exosomes prepared from the *HTB9* tumor cells were more concentrated in the liver and spleen with gradually increased intensity from 48 h to 72 h post injection. In contrast, non-malignant *HEK293*-derived exosomes tend to spread from liver to lung and spleen after 48 h post injection. Using ultracentrifugation isolated *HEK293* exosomes for in vivo biodistribution study in *C57BL/6* mice has been reported[62], which displayed the similar biodistribution profile as we observed using NanoPom isolated *HEK293* exosomes in Fig. 6. Although *HTB9* exosome biodistribution profile has not been reported elsewhere previously using ultracentrifugation isolation method, our NanoPom isolated *HTB9* exosomes did show distinguishable profile than that from *HEK293* exosomes, which implies the different exosome types derived from different parent cell lines. Figure 6c provides the repetitive and quantitative analysis of biodistribution pattern over time. The results potentially indicate the distinctive biodistribution profile from cancer-associated exosomes which could be very important for understanding tumor cell-mediated communications within the microenvironment. Currently, substantial efforts have been made for using exosomes as therapeutic agents or delivery vehicle in vivo. Thus, being able to reproducibly prepare pure and homogenous exosomes is critical for maintaining consistent biodistribution patterns. During the entire in vivo study, we did not observe any adverse effect and NanoPoms prepared exosomes are well tolerated.

## Discussion

All living cells secret EVs which are diverse populations with heterogeneous molecular functions[63–65]. Recently substantial researches have shown the heterogeneity of EVs[21,66–70] in terms of density, molecular cargos, and morphology, which are even released by a single cell type[16,17,71]. Our recent study also observed that molecular packaging of secreted EVs or exosomes is highly variable upon the change of cellular culture environment as well as surrounding community[72]. Thus, the more advanced analytical methods are urgently needed to be able to decipher such heterogeneity in precision. Additionally, for therapeutic delivery, the well-defined molecular components from the homogenous exosome population are also critical to precisely maintain controllable biodistribution pattern and delivery behavior[73]. We also demonstrated that NanoPoms method is applicable to nearly all types of biological fluids, including human blood, urine, cow's milk, and cell culture medium, etc. (Supplementary Fig. s6). The isolation process is simple without need of ultracentrifugation. As shown in Supplementary Fig. s6, the uniform exosomes can be captured on bead surface with antibody defined specificity, which is much purer than ultracentrifugation isolated EVs with co-purified membrane debris shown in the TEM image. The entire isolation protocol is straightforward and cost-effective, amenable for scaling up, sterilization settings, and Good Manufacturing Practice (GMP) operations (see Supplementary Table s4). Due to the unique 3D nano-scale pom poms structure and specific marker defined capture-release process, our developed isolation approach can prepare homogenous exosome subpopulations which enrich tumor associated biomarkers. In our study, the NGS and ddPCR analysis demonstrated that DNAs isolated from NanoPoms prepared exosomes are enriched for tumor-associated DNA mutations which are highly relevant to the bladder cancer. This evidence further supports that specific cancer-associated biomarker are enriched in exosome type urinary EVs and can serve as surrogates for tumor cells.

The miRNAs represent the most dynamic nucleic acid cargos in exosomes, which is relatively sensitive to external stimulus and

changes. Thus, in order to gauge the impact of light release process on exosome isolation via NanoPoms approach, we compared miRNA profiles with or without light release process, which did not show statistically significant differences based on dendrogram clustering analysis (Fig. 4a and Supplementary Fig. s5). The light release process also is able to ensure the specificity via releasing captured exosomes only, to avoid non-specific binders. This data supports the quality and integrity of NanoPoms prepared exosomes as a novel, rapid, and easy-to-use method. Currently, although urinary miRNA profiling is highly essential for BC diagnosis, such study and relevant database have not been fully established yet. NanoPoms based exosome sample preparation could potentially speed up this research direction by offering much simple and specific exosome preparation.

The urinary exosome cargos at the protein level from our study reveals the consistent expression of exosomal proteins *CD9*, *CD63*, and *TSG101* from both patient urinary exosomes and cell lines using UC and NanoPoms preparations. In contrast, *EDIL-3* levels have been observed much higher in BC patient urinary exosomes compared to healthy individuals which is consistent with reported literature[56], indicating the high-quality preparation of exosome using NanoPoms approach (Fig. 5a). Further, the proteomic profiling also supports that NanoPoms prepared urinary exosome proteins can be used to differentiate BC disease from healthy status (Fig. 5b and c, and Supplementary Table s3) with unique identification of pathogenesis relevant exosomes proteins, suggesting a promising avenue using NanoPoms prepared exosomes to develop non-invasive bladder cancer diagnosis. Overall, we identified 10 more miRNAs and 11 more proteins which are uniquely and highly expressed only from the BC patient using NanoPoms preparation, which potentially expanded the landscape of targetable cancer biomarkers.

In order to further prove the integrity and biological activity of NanoPoms prepared exosomes, in vivo biodistribution study exhibits distinctive distribution patterns between tumor-associated exosomes and non- malignant exosomes (Fig. 6). This result may indicate that different subtypes and sources of exosomes could have impact on the performance of drug delivery while using exosomes as the carrier. To date, the therapeutic potential of different subpopulations of exosomes is not well known. It has been discussed that possibly only a small fraction of the exosomes from a cell can mediate the therapeutic effects[73]. Thus, the reproducible isolation of specific exosome subpopulations is essential to support the development of exosome-based therapeutic delivery. The specific isolation and enrichment of exosome subtypes enabled by NanoPoms approach with marker definition could open an avenue for preparing pure and homogenous exosomes with improved therapeutic efficacy.

## Methods

**Fabrication and characterization of nano pom-poms**. The proprietary bead fabrication follows the protocol of $Fe_3O_4/SiO_2$ core-shell-based particle method with surface anchored graphene oxide nanosheets via carboxamide covalent bonds and EDC/NHS chemistry, and further modified with (3-aminopropyl) triethoxysilane (APTES), polydopamine, and streptavidin (Vector Laboratories, USA, SA-5000). Beads were washed with Phosphate Buffered Saline with Tween® 20 (PBST) then resuspend in 1 ml PBST and 0.09% $NaN_3$ solution for storage at 4 °C. In this study we used the pan capture with a mixture of *CD9*, *CD63*, and *CD81* antibodies for bead-conjugation. For in vivo biodistribution study, we used *CD9* antibody conjugated NanoPoms to prepare *HTB9* and *HEK* cells derived exosomes. After bead fabrication and conjugation, XPS analysis was used (PHI 5000 VERSA PROBE II, USA) with an Al anode of the x-ray source (46.95 eV) and 100 µ X-ray beam size for operating at 23.2 W. The power of the source was reduced to minimize X-ray damage for analyzing exosomes on bead surface.

The EV isolation from patients' plasma, urine, or cow milk and conditioned cell culture media were performed by incubation of 100 µL antibody-beads complex with 1 mL of samples at 4 °C overnight. After washing, the photorelease was performed using Analytikjena UVP 2UV Transilluminator Plus at 365 nm wavelength at 4 °C for 15 min (~6 mW/cm2). The UC isolation of EVs followed the

well-documented protocols published previous[72]. Briefly, to remove any possible apoptotic bodies and large cell debris, the supernatants were centrifuged at 10,000 g for 30 mins, then transferred to ultracentrifuge tube (Thermo Scientific, USA) for ultracentrifugation at 100,000 g for 70 min (Sorvall™ MTX150 Micro-Ultracentrifuge, USA), with second ultracentrifugation (100,000 g for 70 min) for finally collecting EV pellets. The size characterization of EVs was performed using the nanoparticle tracking analysis (NTA) Nano-Sight LM10 (Malvern Panalytical, United Kingdom). Post-acquisition parameters were adjusted to a screen gain of 10.0 and a detection threshold to 5. Standard 100 nm nanoparticles were used for calibration. Appropriate sample dilution in 1× PBS (Phosphate Buffered Saline) was evaluated before every measurement with five repeats for each measurement.

**Exosome DNA extraction and NGS sequencing**. Both bladder cancer patient urine and healthy urine samples were obtained from the University of Kansas Cancer Center's Biospecimen Repository. The KU cancer center biospecimen repository frequently stores banked human patient samples following the approved IRB protocols. Frozen urine samples were thawed overnight at 4 °C and pre-centrifuged at 4 °C 10,000 g for 30 min to remove cell debris. By using NanoPoms isolation, the extracted exosomes were treated with DNase I before DNA extraction. The QIAamp DNA Mini Kit (Qiagen, USA, cat. no. 51304) was utilized to extract DNA from all EV samples. The addition of 1 µL of an aqueous solution containing 10 µg of carrier DNA (poly dA) to 200 µL Buffer AL was used to ensure binding conditions are optimal for low copy number DNA according to the manufacturer's protocols. DNA was Eluted in 20 µL Buffer AE. DNA concentrations were measured using a Nanodrop platform at an absorbance at 260 and 280 nm subtracted by the background value of carrier ploy dA only.

The library preparation by targeted enrichment using Qiagen GeneRead QIAact AIT DNA UMI and GeneRead clonal Amp Q Kits, was subjected to next-generation sequencing (NGS) to generate FASTQ files (text-based format for storing nucleotide sequences). This test is a targeted NGS Panel that encompasses 30 genes and 1411 variants (*AKT1, ALK1, BRAF, CTNNB1, DDR2, EGFR, ERBB2, ERBB3, ERBB4, ESR1, FBXW7, FGFR1, FGFR2, FGFR3, FLT3, GNA11, GNAQ, HRAS, KIT, KRAS, MAP2K1, MAP2K2, MET, NOTCH1, NRAS, PDGFRA, PIK3CA, RAF1, SMAD4, STK11*) with variable full exon or partial region. The reads are mapped to the Homo_sapiens_sequence hg19 reference and variants identified using QIAGEN QCI-Analyze pipeline.

The extracted DNAs were amplified by PCR to detect the *EGFR* (P00533:p. Thr790Met) mutation. The sequences of primers for PCR were as follows: Primer F, 5′-ATGCGTCTTCACCTGGAA-3′; primer R, 5′-ATCCTGGCTCCTTATCT CC-3′. Primers were designed by Primer3Plus online (https://www.bioinformatics.nl/cgi-bin/primer3plus/primer3plus.cgi). The PCR assay was performed with Promega GoTaq Flexi DNA Polymerase kit in a 50-µL mixture containing 10 µL of 5× PCR buffer, 0.25 µL GoTaq Flexi DNA Polymerase, 10 µM of each primer (IDT, USA) and 20 µL of DNA in an ABI PCR instrument (Applied Biosystems, USA). The PCR conditions were as follows: Initial denaturation at 95°C for 2 min, followed by 35 cycles at 95°C for 15 sec, 54°C for 30 sec and 72°C for 40 sec, then a hold at 72°C for 5 min and a final permanent hold at 4°C. The 319 bp DNA size of PCR products were clarified by 1% agarose gel electrophoresis using 5 µL PCR products and remained DNA were purified by QIAquick PCR Purification Kit (Qiagen, USA, cat no. 28104). The purified PCR products were sequenced by Sanger Sequencing approach (GeneWiz, USA) using the same primers above.

**Exosome RNA extraction and NGS sequencing**. The miRNeasy Mini Kit (Qiagen, USA, cat no. 217004) was used to extract total RNA from all EV samples per manufacture's protocols. The amount of 700 µL QIAzol lysis reagent was adapted according to the manual. To achieve a higher RNA yield, the first eluate of 30 µL was applied to the membrane a second time. Isolated RNAs were quantified by High Sensitivity RNA ScreenTape Assay using Agilent TapeStation 2200 (Agilent, USA, 5067-5579, 5067-5580). Total RNA was stored at −80 °C until small RNA Library preparation. The QIAseq miRNA Library is prepared for Single Read 75 bp sequencing, with UMI tag per manufacture's protocols. After small RNA sequencing using Illumina MiSeq system, the Qiagen specific UMI analysis per the kit instruction was performed with details in supplementary information.

**Droplet digital PCR**. A pair of probes and a pair of primers were designed to detect *EGFR* and *TERT* mutation respectively. Due to the short size of the probe, in order to increase the hybridization properties and melting temperature, Locked Nucleic Acid (LNA) bases were introduced on the bases indicated with a "+". One probe was designed to recognize wildtype (5′-TET/T + CATC + A + C + GC/ZEN/ A + GCTC/-3′ IABkFQ). The second probe was designed to recognize the *EGFR* (P00533:p.Thr790Met) mutation loci, (5′-6FAM/T + CATC + A + T + GC/ZEN/ A + GC + TC/-3′ IABkFQ). Primers were designed to cover both side of detection loci. For *TERT*, a probe was designed to detect both C228T and C250T mutation as both mutations result in the same sequencing string[74], with (*TERT* Mut:/56-FAM/ CCC + C + T + T + CCGG/3IABkFQ/). A second probe was designed to recognize the C228 loci, also containing LNA bases, (TERT WT, /5HEX/ CCCC + C + T + CCGG/3IABkFQ/). Probes and primers were custom synthesized by Integrated DNA Technologies (IDT). Amplifications were performed in a 20 µL reaction containing 1 × ddPCR Supermix for Probes (No dUTP), (Bio-Rad, USA,

cat no. 1863024), 250 nM of probes and 900 nM of primers and 8 µL exosome DNA template. Droplets were generated using the QX200 AutoDG Droplet Digtal PCR System (Bio-Rad). Droplets were transferred to a 96-well plate for PCR amplification in the QX200 Droplet Reader. Amplifications were performed using the following cycling conditions: 1 cycle of 95 °C for 10 min, then 40 cycles of 94 °C for 30 s and 60 °C for 1 min, followed by 1 cycle of 98 °C for 10 min for enzyme deactivation. Keep all ramp rate at 2 °C/sec. QuantaSoft analysis software (Bio-Rad, USA) was used to acquire and analyze data.

**Western blotting and proteomic analysis.** The 5 mL of each urine sample for two patients and one healthy control were used for exosome isolation and subsequent Western blot analysis. 40 mL of HTB-9 conditional cell culture media and 40 mg cell pellets were also used as controls in this study. Samples were lysed in 1× RIPA buffer supplemented with protease inhibitors for 15 min on ice. Only cell sample were ultrasonicated for 1 min. Protein concentration was quantified using Micro BCA (Bicinchoninic acid) Protein Assay Kit (Thermo Fisher, 23235). The absorbances were read at 562 nm on a Synergy H1 reader (BioTek, USA). All sample concentration were adjusted to 0.1 µg/µL. Western blotting was performed under reducing conditions (RIPA buffer, β-mercaptoethanol and Halt Protease Inhibitor Cocktail, EDTA-Free) at 95 °C for 5 min. 20 µL of protein lysate, each, were loaded onto 4-20% Mini-PROTEAN TGX Precast Protein Gels (BioRad, USA, 4561093). The separated proteins were transferred to a PVDF membrane (BioRad, USA, 1620218). After blocking the membrane in Intercept (PBS) Blocking Buffer (LI-COR, 927-70001) for one hour at room temperature, it was incubated over-night with the primary antibody at 4 °C, followed by another incubation with the secondary antibody for half hour at room temperature. The following primary antibodies were used, all diluted in blocking buffer (1:1000): anti-CD9 (Thermo Fisher, 10626D), anti-CD63 (Thermo Fisher, USA, 10626D), anti-EDIL3 (Abcam, ab88667), anti-MUC4 (Abcam, ab60720), anti-TSG101 (Invitrogen, USA, PA5-86445), anti-ANXA7 (LSBio, USA, LS-C387129-100). The secondary anti-mouse and anti-rabbit IRDye 800CW antibodies (LI-COR, USA, 926-32210 and 926-32211) were applied in 1:15,000 dilution. Imaging were performed by LI-COR Odyssey CLx system.

Urinary EV pellets resultant from ~2 mL of urine from both bladder cancer patients and healthy individuals (four biological replications for each group) were reconstituted in 400 µL of M-PER Mammalian Protein Extraction Buffer supplemented with 1× Halt Protease Inhibitors and sonicated in an ultrasonic water bath for 15 min. NanoPom exosome samples were processed by in-gel digestion for LC-MS analysis. Approximately 20 µg of protein from each sample was resolved on a 4-20% gradient gel (Bio-Rad), then visualized by SimplyBlue SafeStain (ThermoFisher). Each lane was cut into 8 slices of approximately the same size, then reduced, alkylated, and digested with 400 ng of trypsin overnight at 37 °C. Digestion was quenched with 1% formic acid (FA) in 50 mM ammonium bicarbonate buffer/50% acetonitrile (ACN). Peptides were dried using a speed-vac and stored at −20 °C. For LC-MS analysis, peptides were reconstituted in 3% ACN/0.1% FA. Peptides were injected onto an Acclaim PepMap 100 C18 trap column (75 µm x 2 cm, ThermoFisher) using an Agilent 1260 Infinity capillary pump and auto sampler (Agilent Technologies). The autosampler was maintained at 4 °C, the capillary pump flow rate was set to 1.5 µL/min, and an isocratic solvent system consisting of 3% ACN/0.1% FA. After 10 min, the trap column was valve was switched to be in-line with an Acclaim PepMap RSLC C18 analytical column (50 µm x 25 cm, ThermoFisher), using an Agilent 1290 Infinity II column compartment, kept at 42 °C. Peptides were resolved on the analytical column using an Agilent 1290 Infinity II UHPLC with nanoflow passive split flow adapter, maintaining 200 µL/min flow pre-split, resulting in ~300 nL/min flow on the analytical column at the beginning of the run. A two solvent system consisting of (A) water/0.1% FA and (B) ACN/0.1% FA was used, with a gradient as follows: 3% B at 0 min, ramping to 8% B at 8 min, ramping to 26% B at 90 min, ramping to 35% B at 105 min, ramping to 40% B at 120 min, then ramping to 70% B at 122, held at 70% B until 127 min, before returning to 3% at 130 min and holding until the end of the run at 150 min, with a post run equilibration of 12 min. Eluted peptides were analyzed by an Agilent 6550 QToF mass spectrometer equipped with a G1992A nanoESI source (Agilent Technologies). The source parameters were as follows: drying gas temperature was set to 200 °C, flow of 11 L/min, a capillary voltage of 1200 V, and fragmentor voltage of 360 V was used. Data was acquired in positive ion mode using data dependent acquisition, with an MS scan range of 290 to 1700 m/z at 8 spectra/s, MS/MS scan range of 50-1700 m/z at 3 spectra/s, and an isolation width set to narrow (~1.3 m/z). Maximum precursors per cycle was set to 10, with dynamic exclusion enabled after 2 spectra, and released time set to 0.5 min. Peptides were fragmented by collision induced dissociation (CID) using N2 gas and a variable collision energy depending on the precursor charge and m/z. Reference mass correction in real time was enabled, with lock masses of 299 and 1221 m/z used. Data acquired for each sample was converted to Mascot Generic Format (.MGF) using the Agilent Data Reprocessor (Agilent Technologies). Database searching of .MGF files was done using Mascot Daemon v2.2.2 (Matrix Science). Data was searched against a concatenated decoy FASTA file containing Homo sapiens proteins downloaded from Uniprot. Search results from all 3 engines was combined and analyzed using Scaffold4 v4.8.1 (Proteome Software Inc.). Thresholds of 1% FDR protein, 1% FDR peptide, and 2 peptides minimum were set for protein identification.

**SEM and TEM.** exosome- bead particle complex was resuspended in 200 µL cold PBS solution. For electron microscope evaluation, exosome-particle complexes were washed with pure water followed by the fixation in a 2% EMS-quality paraformaldehyde aqueous solution. 5 µL of exosome-particle mixtures were added to cleaned silicon chips and immobilized after drying EVs under a ventilation hood. Samples on silicon chips were mounted on a SEM stage by carbon paste. A coating of gold-palladium alloy was applied to improve SEM image background. SEM was performed under low beam energies (7 kV) on Hitachi SU8230 filed emission scanning electron microscope. For TEM, ~5 µL of each exosome-particle complex was left to adhere onto formvar carbon coated copper Grid 200 mesh (Electron Microscopy Sciences) for 5 mins followed by 5 mins of negative staining with 2% aqueous uranyl acetate. For immune staining TEM, the antiCD63 gold nanoparticles (~10 nm) were used to bind to CD9 bead-captured exosomes. Excess liquids were blotted by filter papers. Total grid preparation was performed at room temperature till totally air-dried under a ventilation hood for 25 mins. Images were acquired on the same day at 75 kV using Hitachi H-8100 transmission electron microscope.

**In vivo biodistribution analysis.** The human bladder cancer cell line HTB-9 (ATCC, USA, 5637) and the negative control of human embryonic kidney epithelial cell line HEK293(ATCC, USA, CRL-1573) were cultured in DMEM and MEM respectively, supplemented with 10% normal FBS and 1% penicillin/streptomycin. Once the cell cultures reached ~70% confluency, the media was replaced with fresh media containing 10% exosome-depleted FBS (Thermo Fisher, USA, A2720803). The cells were cultured for an additional 72 h before the conditioned media were collected. exosomes were isolated using NanoPoms approach and subsequently incubated with 1 mM fluorescent lipophilic tracer DiR (1,1-dioctadecyl-3,3,3,3-tetramethylindotricarbocyanine iodide) (Invitrogen, USA, D12731) at room temperature (RT) for 15 min. DiR-labeled exosomes or free DiR dyes were segregated using Amicon Ultra-15 Centrifugal Filter method. The $2.0 \times 10^9$ particles/ml of isolated exosomes measured via NTA were used for each mouse injection. The 6- to 8-week-old female BALB/cJ mice were used. The animal IACUC protocols have been approved by the University of Kansas Institutional Animal Care and Use Committee with protocol number 258-01 and operated in the KU Animal Care Unit. Freshly purified DiR-labeled exosomes were injected through the tail vein for intravenous (i.v.) injection. The In-Vivo Systems (Bruker, USA) with high-sensitive CCD camera was used for collecting fluorescence, luminescence and X-ray images. Isoflurane sedated live mice were taken fluorescence and X-ray images prior to the animals were sacrificed, then main organs (brain, heart, lung, liver, kidney and spleen) were harvested for fluorescence imaging in 3 mins (excitation 730 nm, emission 790 nm), X-ray imaging (120 mm FOV, 1 min) and luminescence imaging (90 fov, 0.2 sec) at 24 h, 48 h and 72 h time points, respectively. The data were analyzed using the Bruker MI software.

**Statistics and reproducibility.** All statistical tests were performed under the open-source statistics using GraphPad Prism software 8 (San Diego, CA), including heatmap, ROC analysis, and clustering analysis. The one-way ANOVA and t test were used. Differences are considered statistically significant at $P < 0.05$. *$P < 0.05$; **$P < 0.01$; ***$P < 0.001$. The Venn diagram was analyzed using open software from the Bioinformatics & Evolutionary Genomics. The bioinformatic analysis of small RNAs was detailed in supplementary Methods.

**Reporting summary.** Further information on research design is available in the Nature Research Reporting Summary linked to this article.

## Data availability

All data generated or analyzed during this study are included in this published article (and its supplementary information file). Source data for the graphs and charts in the main figures is provided in the supplementary data files and any remaining information can be obtained from the corresponding author upon reasonable request. The sequence data are deposited in the Genbank nucleotide sequence database under Accession: PRJNA799354 with ID: 799354. The mass spectrometry proteomics data have been deposited to the ProteomeXchange Consortium via the PRIDE partner repository with the dataset identifier PXD034454 and 10.6019/PXD034454.

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

## Acknowledgements

We thank Xinbao Hao for helping with the in vivo biodistribution imaging protocols. We also thank the assistant from Jennifer Hackett from the Genomic Core at the University of Kansas for library preparation, quality check and next-generation sequencing. We thank Drs. T. Hamerly and R. Dinglasan for provision of mass spectrometry and bioinformatic support, leveraging resources provided to their laboratory by the University of Florida Office of the Vice President of Research, the College of Veterinary Medicine, and the Emerging Pathogens Institute. We acknowledge the support of The University of Kansas Cancer Center's Biospecimen Repository Core Facility staff, funded in part by the National Cancer Institute Cancer Center Support Grant P30 CA168524 (A.K.G.). This work is supported by the NIH NIGMS MIRA award 1R35GM133794 to MH and the NIH NCI R43 CA221536-01A1 to MH and Clara Biotech Inc.

## Author contributions

Conceptualization: M.H., Y.Z. Methodology: N.H., S.T., C.Z., Z.G., L.X., Z.P., A.K.G. Investigation: N.H., Z.G. Visualization: M.H., S.T., C.Z. Supervision: M.H., Y.Z., C.Z., L.X., A.K.G. Writing: M.H., Y.Z., N.H., A.K.G.

## Competing interests

The authors declare the following competing interests: [Author M.H. has patent application: Methods for generative therapeutic delivery platform (PCT/US2019/057237) and patent application: Capture and Photorelease of extracellular vesicles and exosomes (US 63/148,781) licensed by Clara Biotech Inc]. The remaining authors declare no competing interests.
