## [Peer Review File · Communications Biology]

Reviewers' comments:

Reviewer #1 (Remarks to the Author):

The study of exosomes is increasingly useful both for the determination of biomarkers for the diagnosis and monitoring of pathologies and for their use as pharmacological vectors. The availability of precise methods for the isolation of exosomes is one of the limitations we have to handle these particles. This paper presents an interesting method to increase the capacity to isolate exosomes as pure as possible without dragging contaminants from other vesicles. The paper is well structured and developed, but despite this it needs major corrections to be published. The comments derived from the review are described below.

Comments

Major revisions

To clarify the TEM and SEM technology, it is not very clear in the text what it is used for, although it is described in materials and methods, a clarification on this point would be interesting. It is an important point to be able to properly assess the results obtained.

In line 77: It talks about obtaining exosomes from tumor tissue, but in the text there is no reference to studies with these particles. Furthermore, exosomes are exocytosis particles that are detected in biological liquids and not in tissues. Review.

Using the NanoPumps method, were any pathological variants identified in the control group other than PDGFRA? Describe in the text.

Figure 2: We used 11 patients with bladder cancer versus only 4 control patients. What is the reason for this difference in group size? In addition, the text indicates that 56.8% of the controls by the ultracentrifugation method had the PDGFRA variant, whereas in Figure 2A, only 1 of 4 controls expressed this variant (25%), how can this difference be justified? Could this be an outlier result of NGS?

Figure 2C: Why the EGFR and TERT genes were selected for this study, they do not appear in the gene matrix of sections 2A and 2B. Why is there a difference in the size of the healthy and diseased groups between the study in Figure 2C and 2A-B?

Figure 3A: Increase the quality of the images, the dots are not well visible in the circles.

Figure 5: In the western blot it would be necessary to add a marker gene (for example tubulin) to verify that the observed differences are not due to the fact that different amounts of protein have been loaded in each lane, is this done? DNA and RNA studies have been done with the HTB-9 cells. It could be interesting to see if they are shared with the patients with bladder cancer.

Figure 6: This experiment has been performed with exosomes obtained by other methods such as ultracentrifugation, and differences are observed. This could be an interesting comparison as has been done throughout the paper.

It would be interesting to reduce the results section and expand the discussion. The discussion is a bit poor.

The text could include data from studies carried out in other biological fluids, not only in urine. It would allow us to see if the behavior is the same or if there are differences between the matrices. It would enrich the article.

The authors do not refer to the permission of the ethics committee necessary to perform studies with patient samples, do they have this permission? Reference it.

Other comments

Line 30-31: Give an example of FDA-approved markers. It would facilitate the reader's understanding.

Lines 120-123: This sentence "This NanoPoms method is applicable to nearly all types of biological fluids, including human blood, urine, cow's milk, and cell culture medium, etc. (Fig. S3). The operation protocol is simple and cost-effective, amenable for scaling up, sterilization settings, and GMP operations (see Table S1)", seems to me to be more appropriate for discussion.

Line 133: PDGFRA variant, is it a gene? And if it really is a variant of what gene? Review and correct

Line 158 and 160: a.u.c capitalized.

Line 164: Fig 2c, with the c in capital letters.

Line 186-187: In the sentence "As expected, the EGFR heterozygosity was not detected from wide-type control WBCs from matched patients", rewrite leads to confusion of the term wide-type because the exosome and WBC samples are derived from the same patient?

Line 208: Correct "exosome miRANs".

Line 338: Correct "nano pom poms".

Line 384: Describe "PBST".

Line 387: Describe "XPS".

Line 401: "Malvern Panalytical" add country

Line 403: Describe "PBS".

Line 409: Complete reference "Qiagen, 51304"

Line 425: Reference "Primer3Plus online"

Throughout the material and methods section, commercial product references should be completed with the company name and country.

Clarify other abbreviations: XPS, BCA protein, GMP

Table S1: In the cost table, it is indicated that NanoPoms does not require instrumentation, but an ultracentrifugation step is performed?

Reviewer #2 (Remarks to the Author):

The manuscript entitled "Nano Pom-poms Prepared Exosomes for Highly Specific Cancer Biomarker Detection" shows that 3D-structured nanographene immunomagnetic particles with flower pom-poms morphology and photo-click chemistry can be applied for specific marker-defined capture and release of exosomes from several types of biological fluids. Authors used this specific exosome isolation approach for identification of targetable cancer biomarkers such as multi-omic exosome analysis of bladder cancer patient tissue fluids using NGS analysis of somatic DNA mutations, miRNAs, and the global proteome. Given that this new approach provide facile tool for isolation of EVs, this study is likely to attract attention in the field for the EV studies. I support publication of this manuscript with a condition that manuscript needs to be more proofed to abide by journal's criteria.

POINT-TO-POINT RESPONSES TO REVIEWERS

Reviewer #1:

The study of exosomes is increasingly useful both for the determination of biomarkers for the diagnosis and monitoring of pathologies and for their use as pharmacological vectors. The availability of precise methods for the isolation of exosomes is one of the limitations we have to handle these particles. This paper presents an interesting method to increase the capacity to isolate exosomes as pure as possible without dragging contaminants from other vesicles. The paper is well structured and developed, but despite this it needs major corrections to be published. The comments derived from the review are described below.

Comments: Major revisions

To clarify the TEM and SEM technology, it is not very clear in the text what it is used for, although it is described in materials and methods, a clarification on this point would be interesting. It is an important point to be able to properly assess the results obtained.

We appreciate reviewer's positive recognition and careful review. To follow reviewer's suggestion, we added more description in line 109 as "Transmission electron microscope (TEM) cross sectional view illustrates the plumose surface morphology, which is further confirmed by the scanning electron microscope (SEM) top view showing the greatly enhanced layers of surface area. The immune gold nanoparticle staining TEM imaging utilized antiCD63 gold nanoparticles (~10 nm) to double confirm the surface captured exosomes."

In line 77: It talks about obtaining exosomes from tumor tissue, but in the text there is no reference to studies with these particles. Furthermore, exosomes are exocytosis particles that are detected in biological liquids and not in tissues. Review.

For improving the clarity, we follow reviewer's suggestion to make changes in line 77 to indicate the comparative study using tumor tissues (not exosomes extracted from tissue), as below: "In this paper, we demonstrated such capability for analyzing exosomes derived from bladder cancer patient tissue fluids including urine and plasma, and compared with tumor tissues by the next generation sequencing (NGS) of somatic DNA mutations, miRNAs, and the global proteome, for achieving non-invasive, ultra-sensitive diagnosis of bladder cancer."

Using the NanoPomps method, were any pathological variants identified in the control group other than PDGFRA? Describe in the text.

Using the NanoPomps method, there is no any pathological variants identified in the control group as shown in Fig. 2A. In contrast, UC method led to one PDGFRA pathological variant identified in the control group. We follow reviewer's suggestion and add more description to clarify in the line 136: "NanoPoms prepared healthy control samples did not show any pathological variants."

Figure 2: We used 11 patients with bladder cancer versus only 4 control patients. What is the reason for this difference in group size? In addition, the text indicates that 56.8% of the controls

by the ultracentrifugation method had the PDGFRA variant, whereas in Figure 2A, only 1 of 4 controls expressed this variant (25%), how can this difference be justified? Could this be an outlier result of NGS?

We appreciate reviewer's careful review. The typical 2:1 ratio between disease and control groups were used here following statistical power analysis, thus, the healthy control samples were less than bladder cancer patient samples. The 56.8% frequency mentioned in the text is the detection frequency of this certain gene mutation from the given sample. In general, such detection frequency > 5% is considered viable and the given sample is surely carrying such mutation. Therefore, this 56.8% detection frequency is not the representation of mutation frequency from entire patient sample group. We added the more details to clarify in line 134.

Figure 2C: Why the EGFR and TERT genes were selected for this study, they do not appear in the gene matrix of sections 2A and 2B. Why is there a difference in the size of the healthy and diseased groups between the study in Figure 2C and 2A-B?

EGFR and TERT genes have been well reported as the biomarkers for diagnosis and prognosis of bladder cancer. So, we selected both for droplet digital PCR analysis from urinary exosomes which is much more sensitive gene detection method (single copy number) than the NGS sequencing. Thus, we detected much more mutation rate in terms of EGFR and TERT even in low frequency from the patient sample group, in contrast to the undetectable result shown in Fig 2A and B using NGS sequencing method. Although NGS method doesn't have comparable detection sensitivity to gene mutations compared to ddPCR, NGS can provide large scale gene panel screening. However, the cost for running NGS gene panel screening is super expensive. We only have \$30,000 budget for this work, thus, half of the samples in Fig. 2C were selected for NGS analysis shown in Fig 2A. We further clarified in the manuscript line 164 to state the detection sensitivity differences between ddPCR and NGS with references support which could explain the EGFR and TERT genes for ddPCR analysis: "While the NGS method provides a broad coverage of interested gene mutation panels in one run, the ddPCR analysis offers much higher detection specificity and sensitivity, and its rapid and low-cost performance is highly attractive for developing clinical cancer diagnostic assays. Thus, we selected EGFR (Thr790Met) and TERT (C228T and C250T) for ddPCR analysis which are well reported as the prognostic and diagnostic markers for bladder cancer³⁸⁻⁴⁰, although not sensitively picked up by NGS analysis in Fig 2A."

Figure 3A: Increase the quality of the images, the dots are not well visible in the circles.

Following reviewer's suggestion, we revised the Fig. 3A color contrast to make it more visible.

Figure 5: In the western blot it would be necessary to add a marker gene (for example tubulin) to verify that the observed differences are not due to the fact that different amounts of protein have been loaded in each lane, is this done? DNA and RNA studies have been done with the HTB-9 cells. It could be interesting to see if they are shared with the patients with bladder cancer.

Thanks for reviewer's suggestion. We tried to have marker protein to serve as the control for western blotting, unfortunately, the marker protein β -actin did not express from all our exosome samples. We used CD9 as the loading control protein which expressed from all our samples

including patient urine exosomes, cells, and cell derived exosomes. The consistent band intensity across all loaded samples indicated the consistent amount of proteins employed in western blotting analysis. In contrast to protein analysis, we did not analyze the HTB-9 cell line extracted DNA and RNA to compare with patient urinary exosome sequencing, because the genetic level is more variable to intra- and extracellular environment especially for micrRNAs. Following reviewer's suggestion, we added more explanation in the line 260: "As shown in Fig. 5A, the Western blot bands from CD9 across all samples exhibited the consistent intensity, indicating the consistent loading amount of employed protein samples, although the β -actin was unable to express from some of our samples as the control."

Figure 6: This experiment has been performed with exosomes obtained by other methods such as ultracentrifugation, and differences are observed. This could be an interesting comparison as has been done throughout the paper.

We appreciate reviewer's interests. Using ultracentrifugation isolated HEK293 exosomes for in vivo biodistribution study in C57BL/6 mice has been done as we cited in reference 61, which displayed the similar biodistribution profile as we observed using NanoPom isolated HEK293 exosomes. Although HTB9 exosome biodistribution profile has not been reported elsewhere previously using ultracentrifugation isolation method, our NanoPom isolated HTB9 exosomes did show distinguishable profile than that from HEK293 exosomes, which could imply the different exosome types derived from different parent cell lines. We added more clarification in line 325 to explain this observation.

It would be interesting to reduce the results section and expand the discussion. The discussion is a bit poor.

We followed reviewer's suggestion and revised the discussion section.

The text could include data from studies carried out in other biological fluids, not only in urine. It would allow us to see if the behavior is the same or if there are differences between the matrices. It would enrich the article.

The text has been added in discussion as reviewer suggested in line 348. We discussed the other biological fluids as we showed in Fig. S6, which proves as the generic isolation method suitable to variable biological samples.

The authors do not refer to the permission of the ethics committee necessary to perform studies with patient samples, do they have this permission? Reference it.

We obtained both bladder cancer patient urine and healthy urine samples from the University of Kansas Cancer Center's Biospecimen Repository. The KU cancer center biospecimen repository frequently stores banked human patient samples following the approved IRB protocols. So, we added such information in manuscript line 414.

Other comments

Line 30-31: Give an example of FDA-approved markers. It would facilitate the reader's understanding.

We followed reviewer's suggestion to add in line 29 and also cited one reference to look for the full list.

Lines 120-123: This sentence "This NanoPoms method is applicable to nearly all types of biological fluids, including human blood, urine, cow's milk, and cell culture medium, etc. (Fig. S3). The operation protocol is simple and cost-effective, amenable for scaling up, sterilization settings, and GMP operations (see Table S1)", seems to me to be more appropriate for discussion.

We followed reviewer's suggestion to move to the discussion.

Line 133: PDGFRA variant, is it a gene? And if it really is a variant of what gene? Review and correct

The PDGFRA is a gene with variant c.1432T>C, p.Ser478Pro. We followed reviewer's suggestion and changed in line 133.

Line 158 and 160: a.u.c capitalized.

We followed reviewer's suggestion and changed in line 158 and 160.

Line 164: Fig 2c, with the c in capital letters.

We followed reviewer's suggestion and changed in line 171.

Line 186-187: In the sentence "As expected, the EGFR heterozygosity was not detected from wide-type control WBCs from matched patients", rewrite leads to confusion of the term wide-type because the exosome and WBC samples are derived from the same patient?

Yes, it is derived from the same patient. We deleted the "wide-type" in line 193 to avoid the confusion.

Line 208: Correct "exosome miRANs".

We appreciate reviewer's careful review. It has been corrected in line 213.

Line 338: Correct "nano pom poms".

We corrected to describe as a nano-scale pom pom morphology.

Line 384: Describe "PBST".

We defined in line 395 as “Phosphate Buffered Saline with Tween® 20 (PBST)”

Line 387: Describe "XPS".

We defined in line 122 as “The X-ray photoelectron spectroscopy (XPS)”

Line 401: "Malvern Panalytical" add country

We added in line 413 as “(Malvern Panalytical, United Kingdom)”

Line 403: Describe "PBS".

We defined in line 415 as “Phosphate Buffered Saline”

Line 409: Complete reference "Qiagen, 51304"

It has been added in line 423 as “(Qiagen, cat. no. 51304)”

Line 425: Reference "Primer3Plus online"

It is public website software, so we added the online link in line 442:
<https://www.bioinformatics.nl/cgi-bin/primer3plus/primer3plus.cgi>

Throughout the material and methods section, commercial product references should be completed with the company name and country.

We followed reviewer’s suggestions and make changes in the manuscript.

Clarify other abbreviations: XPS, BCA protein, GMP

We corrected in lines 122, 389, and 485

Table S1: In the cost table, it is indicated that NanoPoms does not require instrumentation, but an ultracentrifugation step is performed?

The NanoPoms method does not require ultracentrifugation step and this method is superior to ultracentrifugation method.

Reviewer #2 (Remarks to the Author):

The manuscript entitled “Nano Pom-poms Prepared Exosomes for Highly Specific Cancer Biomarker Detection” shows that 3D-structured nanographene immunomagnetic particles with flower pom-poms morphology and photo-click chemistry can be applied for specific marker-defined capture and release of exosomes from several types of biological fluids. Authors used this

specific exosome isolation approach for identification of targetable cancer biomarkers such as multi-omic exosome analysis of bladder cancer patient tissue fluids using NGS analysis of somatic DNA mutations, miRNAs, and the global proteome. Given that this new approach provide facile tool for isolation of EVs, this study is likely to attract attention in the field for the EV studies. I support publication of this manuscript with a condition that manuscript needs to be more proofed to abide by journal's criteria.

We appreciate reviewer's positive recognition of our work. We already did proof reading to comply with journal's criteria.

REVIEWERS' COMMENTS:

Reviewer #3 (Remarks to the Author):

Congratulations to the authors for the great review of the comments they have made, as well as for the excellent work done on the paper.